# YOLOv8A-SD: A Segmentation-Detection Algorithm for Overlooking Scenes in Pig Farms

**DOI:** 10.3390/ani15071000

**Published:** 2025-03-30

**Authors:** Yiran Liao, Yipeng Qiu, Bo Liu, Yibin Qin, Yuchao Wang, Zhijun Wu, Lijia Xu, Ao Feng

**Affiliations:** 1College of Mechanical and Electrical Engineering, Sichuan Agricultural University, Ya’an 625000, China; 18908161812@163.com (Y.L.); ybq142119@163.com (Y.Q.); wangyc0918@yahoo.co.jp (Y.W.); wzj@sicau.edu.cn (Z.W.); fengao@stu.sicau.edu.cn (A.F.); 2College of Information Engineering, Sichuan Agricultural University, Ya’an 625000, China; qiuyipeng@stu.sicau.edu.cn; 3Sichuan Academy of Agricultural Mechanisation Sciences, Ya’an 610000, China; liubo@sicau.edu.cn

**Keywords:** pig identification, YOLOv8, ADown, segmentation detection

## Abstract

Monitoring pigs in large-scale farms through surveillance cameras is challenging due to complex environments and overlapping animals. This study developed an improved computer vision system called YOLOv8A-SD that can accurately count and track pigs from top-view cameras. The system introduces two key improvements: an attention mechanism that helps the model focus on important features of each pig and a practical training strategy that uses original camera footage for training while applying image preprocessing only during testing. This approach achieved highly accurate pig counting (25.05 compared to actual 25.09 pigs) and reliable pig detection (96.1% accuracy). The findings make it easier to implement automated pig monitoring in real farm conditions, as farmers can use raw camera footage for training the system while maintaining high accuracy. This technology provides a practical tool for farmers to monitor pig numbers and distribution automatically, supporting better farm management decisions.

## 1. Introduction

Pork is the predominant meat product consumed globally and holds a significant role in the worldwide supply of animal protein [1]. With the expansion of the global population and the enhancement of living standards, the demand for pork is rising, and total global meat consumption is projected to increase by 63% by 2050 [2]. In this context, Precision Livestock Farming (PLF), as a real-time management system that utilizes technology to automatically monitor livestock, their products, and the farming environment, is of great significance in improving production efficiency and safeguarding animal welfare [3]. However, PLF faces a number of challenges in practical application: first, the efficiency of data collection, as the scale of farming expands while the number of farm personnel decreases, it is difficult to achieve real-time and accurate assessment of each individual by traditional manual monitoring methods; second, the complexity of data management, which requires the establishment of large datasets and the development of automatic classification algorithms through behavioral analyses; in addition, due to economic and cultural constraints and technological infrastructure deficiencies, the integration of new technologies with existing farming operations also poses adaptation challenges [4]. Computer vision technology provides a viable technological path to address PLF challenges through automated visual data acquisition and analysis in animal behavior monitoring, disease prediction, and body weight measurement [5]. As worldwide demand for meat consumption escalates and environmental demands intensify, the animal husbandry sector confronts significant potential and problems related to scaling, standardization, and intelligence transformation [6,7,8,9].

The advancement of large-scale intensive pig farming necessitates real-time health monitoring and individual identification of pigs to enhance production efficiency and animal welfare [10,11,12,13,14]. Electronic ear tagging serves as a non-invasive approach for individual identification, enabling the automated identification and tracking of pigs, hence enhancing breeding efficiency and production management standards. In contrast to conventional manual identification techniques, electronic ear tagging can markedly decrease labor expenses, enhance identification precision, and facilitate real-time monitoring and data acquisition of pigs, therefore offering crucial assistance for precision agriculture [12,13,15,16,17,18,19,20]. Conventional computer vision techniques in pig monitoring mostly employ global threshold segmentation and morphological filtering for image preprocessing [21]. Target localization and form feature extraction through an ellipse fitting technique, together with the development of a dynamic modeling-based weight estimation system [22]. In practice, visible light imaging and explicit statistical modeling have demonstrated efficacy in fulfilling most fundamental monitoring requirements [23]. Nonetheless, these strategies necessitate comprehensive preprocessing and human feature construction in intricate contexts, exhibiting considerable constraints in real-time performance and environmental flexibility [24]. These constraints result in inadequate real-time processing capacity and suboptimal environmental adaptation of the system, necessitating substantial computational resources, which significantly hampers the advancement of intelligence in the pig business [25,26,27].

Recent breakthroughs in deep learning technology provide innovative solutions for pig identification. The target identification algorithms demonstrated by the You Only Look Once (YOLO) series, due to their robust feature extraction capabilities and real-time performance, exhibit significant potential in the domain of intelligent farming [28]. Add among various YOLO-based improvements, PLM-YOLOv5 proposed by Peng et al. achieved a 9% mAP improvement over the baseline YOLOv5 in pig detection by introducing the CBAM attention mechanism and additional detection head, where PLM represents Pig Livestock Monitoring [29]. For multi-object tracking scenarios, YOLOX+BoT-SORT-Slim developed by Tu et al. demonstrated superior performance with 72.24 FPS processing speed, 99.6% MOTA, and 90.2% IDF1 in long-duration (1-h) tests, where BoT-SORT represents Body Feature Enhanced and Transformer-based SORT [30]. Addressing the challenge of low-light environments, Huang et al. proposed IAT-YOLO by combining an illumination-adaptive transformer (IAT) and a coordinate attention mechanism (CA), achieving comprehensive metrics of 96.8% accuracy, 83.8% recall, and 91.8% mAP@0.5 in black pig detection, where IAT represents the illumination-adaptive transformer [31]. These advances demonstrate the potential of attention mechanisms and specialized architectures in livestock monitoring. In parallel developments for poultry farming, Fang et al.’s depth camera-based monitoring system achieved 97.80% Precision and 80.18% Recall (IoU > 0.5) for white chickens, providing valuable insights for cross-species application of computer vision technology [32]. He et al. introduced the PDC-YOLO network by enhancing the YOLOv7 backbone with SPD-Conv structure and AFPN for improved illumination adaptation, achieving 91.97% mAP, 95.11% Precision, and 89.94% Recall with an error rate of 0.002, where PDC stands for Pig Detection and Counting [33]. Lee et al. devised the mixed-ELAN (Efficient Layer Aggregation Network) architecture and implemented it in YOLOv7 and YOLOv9, resulting in a substantial enhancement in the precision of pig behavior recognition [34]. Luo et al. developed PBR-YOLO, an enhanced version of YOLOv8 with GhostNet backbone and efficient multi-scale attention, achieving 82.7% Precision and 78.5% mAP with only 1.23M parameters and 3.9G FLOPs, where PBR represents Piglet Behavior Recognition [35]. Zhong et al. developed YOLO-DLHS-P, which optimizes YOLOv8n through the LAMP pruning algorithm, achieving 52.49% parameter reduction and 79 FPS while improving mAP@0.5 by 1.16%, where DLHS-P denotes Deep Learning High Speed with Pruning [36]. Pu et al. introduced TR-YOLO, enhancing detection accuracy by 1.6 percentage points through the integration of C3DW and C3TR modules without additional computational load, where TR indicates Transformer-based [37]. Li et al. introduced the YOLOv5-KCB algorithm, attaining 98.4% accuracy in pig head and neck recognition and 95.1% in pig face recognition through K-means clustering optimization and CA (Channel Attention) mechanism, where KCB represents K-means Clustering and BiFPN [38].

A systematic literature analysis revealed important knowledge gaps in existing research. The problem of pig-overlap recognition in intensive farming environments, especially when observing from the top viewpoint, remains an urgent challenge. Meanwhile, there are obvious limitations in the adaptability of existing models to the dynamic lighting conditions on farms, which affects the stability of the system in practical applications. In addition, existing studies tend to increase the computational burden while improving the detection accuracy, leading to a decrease in real-time performance, and this balance between accuracy and efficiency needs to be further explored.

To address these challenges, this study presents clear research objectives. From the perspective of cognitive science, it aims to enhance recognition accuracy through attentional mechanisms and dual-task learning strategies; from the perspective of practical value, it is dedicated to solving the key challenges in intensive farming, such as overlap recognition and light adaptation problems. To achieve these goals, this study proposes the YOLOv8A-SD model, which enhances feature extraction through an innovative attention mechanism, adopts a dual-task strategy for segmentation detection to improve performance, and optimizes the design for the top-view scenario of pig farms.

## 2. Materials and Methods

### 2.1. Data Preparation and Preprocessing

The experimental data were obtained from the overhead image of a pig facility in a large-scale farm located in Yucheng District, Ya’an City, Sichuan Province. The Hikvision DS-2CD3T86FWDV2-I3S network camera (Hikvision, Hangzhou, China) (3 million pixels, H.265 encoding, 120 dB wide dynamic range) was used to capture images of porcine behaviors at several times of day and under diverse lighting conditions using camera apparatus positioned at fixed heights of 2.8 m (for training and validation datasets) and 3.5 m (for final test dataset) to ensure the dataset’s robustness for practical uses.

The experimental pigs were Duroc × Landrace × Yorkshire three-way crossbred pigs (DLY), weighing 80–100 kg, with a stocking density of 0.8 head/m^2^. The pig house used traditional concrete flooring. During the feeding period, a standard fattening diet formula was used: 62% corn, 22% soybean meal, 10% wheat bran, and 6% premix (containing Chinese milk vetch extract to enhance heat and humidity resistance), with manual feeding at fixed times daily. All experimental procedures strictly followed animal welfare guidelines, with data collection through environmental cameras causing no stress or interference to the animals.

The dataset for the target detection problem comprises 924 training photos and 216 validation images. The instance segmentation challenge necessitates pixel-level annotation, resulting in a larger dataset comprising 2985 training photos, 1512 validation images, and 278 test images. The augmented dataset significantly enhances the generalization capacity of the segmentation model. To comprehensively evaluate the model’s generalization capability, a completely independent test set was specifically constructed: 220 photos were gathered from a different pig housing environment (camera height 3.5 m) as the conclusive test set. This independent test set features distinct environmental conditions (including lighting and viewing angles) from the training environment, where both segmentation and detection models were trained using data collected from the same pig farm (camera height 2.8 m), enabling better validation of the model’s performance in practical application scenarios. In the experimental design, different processing strategies were used for the training and test data: S-Train denotes training using segmented images (including both training and validation sets), NS-Train denotes training using original images; S-Test denotes testing using the segmented 220 final test images, and NS-Test denotes testing using the original 220 final test images. This design aims to evaluate the effect of different data processing strategies on model performance.

### 2.2. Improved YOLOv8A-SD Model Architecture

YOLOv8A-SD signifies an enhancement of the YOLOv8 base architecture, with ’A’ indicating the integration of the ADown module and ’SD’ indicating that the model possesses dual capabilities for segmentation and detection.

Figure 1 illustrates that the Backbone of the YOLOv8A-SD model, depicted in Figure 1a, comprises five feature extraction layers, P1–P5. Layers P1 and P2 utilize the Conv1 structure, primarily tasked with extracting low-level edge and texture features, while layers P3 to P5 employ the MaxPooled4 structure to progressively extract higher-level semantic information. This hierarchical feature extraction framework significantly enhances the model’s capacity to identify targets across various scales, particularly for detecting small objects in densely populated environments, offering considerable advantages.

The ADown attention module is implemented subsequent to the P4 layer to augment feature expression capacity. This design’s distinctive characteristic is its adaptive spatial attention mechanism. In contrast to the conventional attention module, ADown effectively captures the salient elements of the target region while preserving computing efficiency by integrating spatial downsampling with channel attention. This method is especially effective at addressing pig overlapping issues in top-view scenarios, as it can adaptively modify feature weights to emphasize prominent areas of the target while minimizing background interference.

In Figure 1b, the Neck component employs a multi-scale feature fusion technique, facilitating the efficient integration of diverse feature layers via a feature pyramid network. This architecture optimizes the model’s detection performance for targets of varying scales and improves the semantic representation of the characteristics. Specifically, multi-scale feature fusion offers a more stable and robust feature representation when addressing scale variations and viewpoint distortions in top-down situations.

Figure 1c presents a novel dual-task output architecture for segmentation and detection in the Head portion. This design facilitates the mutual enhancement of the segmentation and detection tasks through a feature-sharing mechanism: the detailed contour information from the segmentation task enhances detection accuracy, while the positional data from the detection task aids the segmentation task in more accurately localizing the target region. Figure 1d illustrates the YOLOv8A-SD model, which possesses dual capabilities of segmentation and detection. This dual-task learning approach not only augments the model’s performance but also amplifies the discriminative quality of the features.

The internal architecture of the ADown module depicted in Figure 1e is illustrated in the lower section of Figure 1, which amplifies the critical region features via the spatial attention mechanism. The module features a lightweight architecture and comprises three essential components: spatial downsampling, channel attention, and residual connectivity. Spatial downsampling decreases computing complexity by lowering feature map resolution. The channel attention mechanism enhances feature representation by discerning correlations across feature channels, and the residual connection guarantees efficient information transfer. This meticulously crafted structure enhances performance while maintaining little computational overhead.

### 2.3. Training Strategy and Implementation

Figure 2 illustrates the comprehensive training and inference procedure of YOLOv8A-SD. The training phase initially employs the segmentation dataset to train the Seg Module, resulting in the acquisition of the distinct segmentation model weights, Seg.pt. This pre-training technique allows the model to acquire more precise target contour information, establishing a foundation for subsequent detection tasks. The segmentation model is trained under pixel-level supervision, utilizing fine-grained labeling information to accurately discern target boundaries, which is crucial for addressing target overlap in congested environments.

The target detection model is trained with a dual parallel technique. NS-Train employs the original image to directly train the Dec Module, resulting in Dec-NS.pt. This method preserves the entirety of the original image’s information, facilitating the model’s understanding of the inherent features and contextual relationships of the target. S-Train analyzes the training picture using Seg.pt prior to training the Dec Module to obtain Dec-S.pt, which emphasizes the target region through segmentation preprocessing and mitigates background interference. This bidirectional training approach aims to maximize the benefits of diverse data types, allowing the model to acquire target properties from several viewpoints.

During the inference phase, the model offers two adaptable testing procedures for unlabeled test datasets. NS-Test employs the original final test images for detection, making it appropriate for situations with constrained processing resources or the necessity for rapid detection. S-Test employs Seg.pt, for instance, segmentation preprocessing before detection, hence augmenting computing overhead, although it can yield more precise detection outcomes, particularly in intricate situations with enhanced stability in performance. This strategic option can be adaptively modified based on the requirements of specific application contexts to achieve a balance between detection precision and processing efficiency.

The adaptable architecture of the training and inference process allows the model to thoroughly assess the influence of various data processing techniques on detection efficacy. By jointly training segmentation and detection tasks, the model enhances its comprehension of the scene’s semantic content and its adaptability to intricate environments.

Table 1 delineates the essential parameter settings for model training to guarantee the reproducibility of the trials. The experiments were conducted on the Windows 11 operating system with PyTorch 2.4.1 deep learning framework and CUDA 12.5 support. The studies utilized NVIDIA GeForce RTX 2080Ti GPUs, employing the Adam optimizer to train for 150 epochs, with an initial learning rate of 0.01, which was dynamically modified using the cosine annealing technique.

### 2.4. Model Evaluation

#### 2.4.1. Metrics

Four fundamental metrics—True Positive (TP), False Positive (FP), True Negative (TN), and False Negative (FN)—are routinely employed to assess model performance in target detection tasks. TP denotes the quantity of accurately identified targets, FP signifies the count of erroneous detections, TN represents the number of correctly disregarded background areas, and FN indicates the number of overlooked targets. These metrics are dimensionless ratios with values ranging from 0 to 1, often expressed as percentages from 0% to 100%, where higher values indicate better model performance.

Precision denotes the accuracy of detection outcomes and signifies the ratio of correct detections to the total number of detection frames. It is computed according to Equation (Equation 1).(1)$$\;\mathrm{Precision}\;=\frac{T\;P}{T\;P+F\;P}\times 100\%$$

Recall reflects the completeness of detection and indicates the proportion of correctly detected in all real targets. Calculated as Equation (Equation 2).(2)$$\;\mathrm{Recall}\;=\frac{T\;P}{T\;P+F\;N}\times 100\%$$

mAP50 is the predominant comprehensive evaluation metric in object detection, representing the mean of the average precision (AP) for each category at an intersection-over-union (IoU) threshold of 0.5. It is computed according to Equation (Equation 3).(3)$$m\;A\;P_{50}=\frac{\sum_{q=1}^{Q}A\;P(q)Q}{\times }100\%$$

IoU (Intersection over Union) is a metric that evaluates the overlap between the predicted segmentation mask and the ground truth mask. It is calculated as Equation (Equation 4).(4)$$\;\mathrm{IoU}\;=\frac{\mathrm{Precision}\times \mathrm{Recall}}{\mathrm{Precision}+\mathrm{Recall}-\mathrm{Precision}\times \mathrm{Recall}}\times 100\%$$

The Dice coefficient, also known as F1-score, is another metric for assessing segmentation accuracy, which is calculated as Equation (Equation 5).(5)$$\;\mathrm{Dice}\;=\frac{2\times \mathrm{Precision}\times \mathrm{Recall}}{\mathrm{Precision}+\mathrm{Recall}}\times 100\%$$

All five evaluation metrics presented above are expressed as percentages, with values ranging from 0% to 100%. Precision, Recall, and mAP50 evaluate detection performance, while IoU and Dice coefficient specifically assess segmentation accuracy. These metrics are consistently presented in percentage form throughout this study, where higher percentages indicate better model performance in their respective aspects of evaluation.

#### 2.4.2. Grad-CAM Heat Map Analysis

The Grad-CAM heat map visually illustrates the model’s focus on various areas of the input image using color gradients. The red and yellow regions in Figure 3b denote the areas of significant focus for the model, primarily concentrated in the pig’s body, particularly around prominent features such as the head and body contour, whereas the blue regions represent the background areas that receive minimal attention. This visualization result demonstrates that the model can precisely identify the target area and reveals a discernible differentiation in the model’s attention levels towards various sections of the target. The distribution patterns of the heat map closely align with the real morphology of the pig, validating the model’s capacity to accurately identify the target’s important traits.

The examination of the heat map confirms that the model’s attention is concentrated on the target item and demonstrates its proficiency in distinguishing overlapping regions, highlighting the model’s supremacy in feature extraction and target recognition.

## 3. Results and Discussion

### 3.1. Comparative Analysis of Indicators

#### 3.1.1. Training Process

Analysis of the experimental results presented in Figure 4 reveals distinct patterns in both confusion matrices and loss trajectories. Through comparative analysis of confusion matrices between NS-Train and S-Train, NS-Train achieves a prediction accuracy of 0.94 for pig categories, while S-Train reaches 0.93. This performance difference reflects the importance of preserving complete target feature information in original images. The texture details, edge contours, and local structural features in raw data provide richer learning materials for the model. The preprocessing segmentation process may introduce additional noise and uncertainty, which affects the model’s learning of original features.

The training process is further characterized by three types of losses, each providing unique insights into model behavior. Regarding box loss (Figure 4c), NS-Train (green line) consistently maintains lower loss values than S-Train (red line), demonstrating a more stable learning process for target localization. This performance advantage stems from more accurate boundary information in original images, enabling the model to better grasp the spatial position features of targets. The dfl loss (Figure 4d) exhibits similar patterns between both approaches, though NS-Train demonstrates slightly more stable convergence. The cls (classification) loss (Figure 4e) reveals that NS-Train achieves lower values throughout training, indicating superior performance in target classification tasks.

From a theoretical standpoint in deep learning, NS-Train’s performance advantage originates from the complete preservation of original scene information. During training, the model directly interacts with unpreprocessed real data distribution, making the feature learning process more natural and complete. The lighting variations, shadow effects, and natural transitions between targets in the original images provide crucial learning cues for the model. This training approach avoids artificial bias and information loss from preprocessing steps, allowing the model to build feature representations closer to actual scenarios. Training with original data enhances the model’s environmental adaptability, enabling better handling of complex situations in real farming scenes, including lighting variations across different periods and varying group density distributions. The consistently lower loss values across all three metrics indicate that training with original images not only simplifies the learning process but also leads to more robust feature representation, enabling the model to better adapt to real-world scenarios where lighting conditions and pig distributions vary significantly.

#### 3.1.2. Final Metrics

The data analysis in Table 2 indicates that each attention module has exceptional performance on the NS-Train dataset. Among these, the Precision metric indicates the accuracy of model recognition, Recall represents the detection rate of the model, and mAP50 provides a complete assessment of the model’s overall performance. The ADown module has superior performance, achieving a Precision of 96.1%, indicating that 96.1% of identified targets are accurate, which translates to fewer than four misclassifications per 100 pigs in a practical agricultural context. This performance is particularly noteworthy when compared with existing YOLO variants: while IAT-YOLO achieved comprehensive metrics of 96.8% accuracy, 83.8% recall, and 91.8% mAP@0.5 in black pig detection, it primarily focused on low-light conditions. PDC-YOLO, despite its sophisticated SPD-Conv structure and AFPN, achieved 91.97% mAP, 95.11% Precision, and 89.94% Recall in standard conditions. The present model maintains comparable or superior performance while handling multiple challenges in complex top-view scenarios. Even compared to YOLOv5-KCB’s impressive 98.4% accuracy in pig head and neck recognition, our model demonstrates strong performance in the more challenging task of whole-body detection under varying conditions.

Alternative attention modules, including AKConv, CAFMAttention, CBAM, and EMA, demonstrated commendable performance, with their Precision consistently exceeding the 0.95 threshold. This consistent performance across different attention mechanisms surpasses PBR-YOLO’s 82.7% Precision and 78.5% mAP despite its use of the GhostNet backbone and efficient multi-scale attention. Similarly, while PLM-YOLOv5 improved mAP by 9% through the CBAM attention mechanism, and TR-YOLO enhanced detection accuracy by 1.6 percentage points through C3DW and C3TR modules, our ADown module exhibits stronger object discrimination capability, particularly in handling overlapping targets.

This level of consistent high performance is crucial for the farm’s daily management, as it guarantees that the monitoring system can consistently deliver trustworthy data to assist managers in staying informed about herd conditions and making precise management decisions.

The original YOLOv8 model, even in the absence of an attention mechanism, demonstrates robust fundamental performance, indicating its inherent proficiency in target recognition. Nevertheless, the incorporation of the attention mechanism markedly enhances the model’s stability in managing intricate scenarios. While YOLOX+BoT-SORT-Slim achieved impressive tracking metrics with 72.24 FPS processing speed and 99.6% MOTA in long-duration tests, and YOLO-DLHS-P achieved 52.49% parameter reduction while improving mAP@0.5 by 1.16%, our focus on environmental adaptability through the attention mechanism provides more robust performance in challenging farming conditions. This enhancement is especially crucial in authentic agricultural settings, where intricate circumstances like variations in lighting and pig congestion frequently arise in pig barns.

The experimental results indicate that performance on the NS-Train dataset is predominantly superior to that on the S-Train dataset. This discovery offers significant insights for practical applications, allowing raw surveillance photos to be utilized directly in model training without intricate preparation. This not only streamlines the system deployment process and diminishes computational resource demands but, more crucially, attains superior recognition outcomes. This advantage arises from the comprehensive information present in the original images, allowing the model to acquire more intricate features and consequently better adjust to the diverse alterations in the actual breeding environment, including variations in lighting conditions between day and night and fluctuations in pig density. These results collectively demonstrate that YOLOv8A-SD achieves a better balance between accuracy and computational efficiency compared to existing approaches while maintaining robust performance across various challenging conditions in real-world farming environments.

### 3.2. Segmentation Model Effect Analysis

Table 3 demonstrates the performance comparison of different attention modules in the segmentation task. From the data, it can be seen that the ADown module achieves optimal performance in several evaluation metrics, with a Precision rate of 93.1%, a Recall rate of 88.6%, an IoU of 83.1%, and a Dice coefficient of 90.8%. This indicates that the ADown module has a significant advantage in target outline extraction and pixel-level classification. The AKConv module follows closely behind in terms of recall rate, slightly better than ADown (88.9%), indicating that it performs well in reducing missed detections. In contrast, the overall performance of the CAFMAttention module is relatively weak, with the lowest IoU (79.5%) and Dice coefficient (88.6%), suggesting that there is still room for improvement in terms of accurate segmentation. The original YOLOv8 model, although it does not have an additional attention mechanism, still shows a good basic performance, with its accuracy (91.7%) and Dice coefficient (89.5%) at a medium level, which confirms the reliability of YOLOv8 as a basic model. Through comparative analysis, it can be found that the introduction of an appropriate attention mechanism can indeed improve the segmentation performance of the model, especially in terms of accuracy rate and boundary localization, and the improvement is more obvious.

The segmentation results in Figure 5 provide a comprehensive visualization of different attention modules’ performance in pig instance segmentation. Each row represents a different attention mechanism’s segmentation results, with two distinct scenarios shown: sparse distribution (left) and clustered distribution (right).

The ADown module (first row) demonstrates the most complete segmentation coverage among all variants. In the sparse scenario, it successfully identifies individual pigs near the pen boundaries and in central areas. However, even ADown shows limitations in the clustered scenario, where some pigs in tight groups are not fully segmented, particularly visible in the central cluster where segmentation boundaries become less distinct.

AKConv (second row) shows similar but slightly reduced performance compared to ADown. While it maintains good detection in clear, unobstructed areas, it struggles more with boundary definition when pigs are in close proximity. This is especially noticeable in the right image, where some pigs in the dense cluster are either partially segmented or missed entirely.

CAFMAttention (third row) exhibits more significant limitations. The visualization reveals clear missed detections, particularly evident in areas where pigs are positioned near pen edges or in unusual poses. The segmentation masks appear less consistent, with more fragmented boundaries and incomplete coverage of pig contours.

CBAM and EMA (fourth and fifth rows) show similar patterns of performance degradation. Both modules have difficulty maintaining consistent segmentation quality across the frame, with notable missed detections in challenging areas such as pen corners and overlapping regions. The blue masks in these cases often appear incomplete or disconnected, especially in the clustered scenario.

The baseline YOLOv8 (bottom row) shows the most conservative segmentation pattern. Its blue masks are less extensive and more fragmented compared to the attention-enhanced variants, with clear gaps in coverage particularly visible in the dense clustering scenario. The comparison clearly demonstrates how each attention mechanism contributes to improved segmentation completeness, though all variants still face challenges in complex scenarios.

### 3.3. Heat Map Visualization and Analysis

Figure 6 depicts the heat map visualization outcomes of three attention mechanism modules. The heat map illustrates the model’s visual attention distribution, employing an analytical approach based on human visual cognition, whereby essential feature areas are prioritized during target observation. The model’s attention distribution pattern is immediately discernible via the color gradients of the heat map, with blue signifying low-attention regions and red denoting high-attention regions.

The ADown module exhibits precise attention distribution characteristics, capable of focusing attention on each individual target. Each hotspot region corresponds to a single pig, demonstrating the model’s ability to effectively distinguish and locate individual animals. From a cognitive principles standpoint, the ADown module has attention distribution features akin to human visual focus, enabling precise localization and concentration on critical areas of the object.

The AKConv module demonstrates a comparable, if somewhat diffused, attention pattern, a characteristic that is beneficial for preserving the perception of target integrity. Meanwhile, the CAFMAttention module utilizes an expansive attention distribution technique, indicative of its inclination to comprehend the target via a synthesis of holistic attributes.

Figure 7 shows the heat map visualization of CBAM, EMA, and the original YOLOv8 model. The CBAM module employs an attention distribution approach that resembles the human cognitive necessity to integrate contextual information for judgment in intricate scenarios, hence enhancing the model’s adaptability to scene alterations, albeit potentially introducing additional contextual data.

The EMA module demonstrates a distinctive attention allocation pattern that equilibrates the processing of local specifics and global characteristics through the dynamic modulation of attention distribution. This balanced approach allows for comprehensive feature extraction while maintaining focus on critical areas.

The original YOLOv8 design exhibits a relatively decentralized attention distribution and lacks an efficient feature selection mechanism. This scenario resembles that of an unseasoned observer who struggles to promptly identify critical feature areas. Through the implementation of attentional processes, the model attains a feature extraction proficiency that aligns more closely with human visual cognition, enabling it to autonomously learn and focus on the most distinguishing visual aspects.

This differential analysis of attentional distribution elucidates the functioning of several attentional processes and offers significant insights into the model’s feature learning process. The various attention modules enhance the model’s feature extraction efficiency through different attention distribution strategies, exhibiting significant advantages in both theoretical and practical applications.

### 3.4. YOLOv8A-SD Segmentation Strategy Analysis

To systematically evaluate the effectiveness of key components in the proposed YOLOv8A-SD framework, ablation experiments were conducted on both the ADown attention mechanism (as shown in Table 2 and Table 3) and the segmentation-detection dual-task strategy. As described in Section 2, the framework employs different combinations of training and testing strategies: NS-Train (training with original images) and S-Train (training with segmented images) for the training phase, NS-Test (testing with original images) and S-Test (testing with segmentation preprocessing) for the inference phase. This section focuses on analyzing the contribution of these different segmentation strategies through comprehensive ablation studies.

Figure 8 presents a comprehensive visualization matrix comparing different training-testing strategies across varying pig density scenarios. The matrix consists of six rows showing original images, segmentation results, and four different training-testing combinations, with each column representing increasing pig density situations.

The first row displays the original surveillance images, showing three typical scenarios: sparse distribution (left), moderate clustering (middle), and dense grouping (right). These raw images highlight the challenging nature of pig detection under different density conditions, with issues such as varying poses, overlapping, and uneven lighting.

The second row presents the segmentation preprocessing results, where blue masks clearly delineate individual pig targets. Through the 6 × 3 visualization matrix, distinct detection patterns emerge across different scenarios. In sparse distribution cases, all strategy combinations demonstrate effective detection performance with minimal missed detections. In dense group scenarios, the model encounters more complex spatial relationships, particularly when multiple pigs cluster tightly together.

The third row (NS-Train and NS-Test) shows detection results with red bounding boxes on the original images. While the detection performs reasonably well in sparse scenarios, some challenges emerge in denser situations, particularly with closely grouped pigs. The fourth row (NS-Train and S-Test) combines original image training with segmentation-aided testing, showing improved detection precision, especially in complex clustering scenarios.

The fifth row (S-Train and NS-Test) reveals the results of training with segmented images but testing on original images. This combination shows certain limitations in dense scenarios, with slightly decreased boundary recognition accuracy. The final row (S-Train and S-Test) demonstrates the full segmentation strategy in both the training and testing phases, showing good individual pig identification but potentially missing some subtle features present in the original images.

Across all density scenarios, the NS-Train and S-Test combination (fourth row) demonstrates particularly robust performance. The red bounding boxes show precise localization while maintaining individual target discrimination, even in challenging, dense scenarios. This combination effectively leverages both the complete feature information from original image training and the enhanced target focus provided by segmentation preprocessing during testing. These observations provide valuable insights for further optimization of model performance in high-density scenarios.

Table 4 presents a comprehensive ablation study to evaluate each component’s contribution. The results can be analyzed from two aspects: model structure and training strategy.

For model structure, adding the ADown attention mechanism significantly improves both detection and segmentation performance. Compared to the baseline YOLOv8, the detection Precision increases from 91.7% to 96.1% (+4.4%), Recall from 87.4% to 90.6% (+3.2%), while maintaining comparable mAP50 (96.3% vs. 96.0%). For segmentation metrics, ADown(S-Train) achieves IoU of 83.1% and Dice of 90.8%, showing improvements of 2.1% and 1.3%, respectively.

Regarding training strategy, the model exhibits different behaviors under various training-testing combinations. When trained with original images (NS-Train), the model achieves better detection performance (Precision: 96.1%, Recall: 90.6%, mAP50: 96.3%) compared to training with segmentation-preprocessed images (S-Train). This suggests that preserving complete visual information during training is beneficial for feature learning. The counting accuracy is also noteworthy, with all configurations achieving close results to the actual value of 25.09 pigs, demonstrating the model’s robustness in dense object scenarios.

The confidence scores provide additional insights into model behavior. YOLOv8A-SD generally shows higher average confidence (0.740–0.783) compared to the baseline (0.714–0.761), indicating improved detection certainty. The wider confidence range in YOLOv8A-SD (e.g., 0.286–0.892) suggests better discrimination between difficult and easy cases.

## 4. Conclusions

This study proposes an enhanced model, YOLOv8A-SD, derived from YOLOv8, to augment feature extraction through the ADown attention mechanism, improve model performance via a dual-task strategy of segmentation and detection, and optimize design for the overlooked scenes of a farm. The experimental findings indicate that YOLOv8A-SD attains a Precision of 96.1% and a mAP50 of 96.3% on the NS-Train dataset, surpassing other attention mechanism modules. The model’s capability to precisely identify the target area and extract relevant features was validated by the Grad-CAM heat map display. The study on training strategies revealed that utilizing a combination of training with raw images and segmenting them during the testing phase (NS-Train and S-Test) achieves optimal performance, with exceptional counting accuracy (25.05 vs. actual 25.09) and the highest average confidence (0.783) among all configurations.

The YOLOv8A-SD model demonstrates effective capability in individual animal recognition, as validated through multiple aspects: the segmentation results show precise delineation of each pig’s contour in standard scenarios, the attention heat maps reveal focused attention distribution to individual animals, and the detection results across different training strategies confirm consistent individual recognition performance under normal conditions. However, error analysis reveals certain limitations, particularly in dense clustering scenarios where boundary recognition accuracy decreases and in cases of unconventional postures where false negatives may occur.

The segmentation model exhibited reliable performance in typical lighting circumstances and moderate overlap scenarios. In comparison to current methodologies, YOLOv8A-SD enhances the model’s feature extraction proficiency through a dual-task learning strategy while maintaining satisfactory detection accuracy under standard conditions. Nevertheless, the model shows limitations in extreme scenarios such as severe overlapping or unusual lighting conditions, indicating room for improvement in environmental adaptability.

In the current implementation stage, farm operators can utilize the system to detect and segment individual pigs in surveillance footage, which helps identify overlapping pigs and facilitates manual counting through visual segmentation results. To use the system, operators need basic familiarity with Anaconda terminal operations, while technical personnel assist with environment setup and hardware deployment. The practical benefits for farms include improved pig counting accuracy through better visual separation of overlapping individuals, enhanced monitoring efficiency, and reduced time spent on manual observation. Farm owners can apply these research results by integrating the detection system into their existing surveillance infrastructure, providing a foundation for semi-automated pig monitoring.

The current system has several limitations that need to be addressed. A more user-friendly interface should be developed to reduce the technical threshold for farm operators. The system’s real-time processing capability needs further optimization for continuous monitoring scenarios. The model’s performance degradation in challenging conditions, such as ultra-high-density farming or severe lighting variations, requires additional attention. Furthermore, the model’s transfer capability to different farm environments and pig breeds shows limitations, suggesting the need for more robust feature learning mechanisms.

Future studies should concentrate on addressing the identified failure cases, particularly in dense population scenarios and extreme environmental conditions. Enhancing the feature extraction network will help improve the model’s segmentation capability for overlapping individuals and unusual postures. To tackle the recognition difficulties of rapidly moving pigs and varying lighting conditions, adaptive feature learning mechanisms need to be developed. The project aims to enhance the target detection head to improve the model’s robustness across different farm environments and pig breeds. Simultaneously, we will investigate lightweighting solutions to diminish computing overhead, enabling real-time monitoring of the model on the farm’s edge devices. These technological advancements can significantly enhance precise pig identification and behavioral analysis in agricultural settings while addressing current limitations.

## Figures and Tables

**Figure 1 animals-15-01000-f001:**
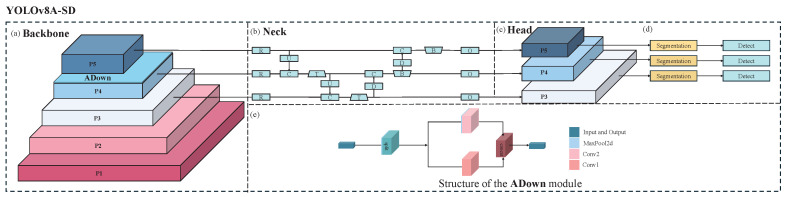
YOLOv8A-SD structure diagram: (**a**) Backbone component showing feature extraction layers P1–P5 with ADown module; (**b**) Neck component showing the feature fusion network; (**c**) Head component showing dual-task architecture for segmentation and detection; (**d**) Overall model structure showing input and output paths; (**e**) Detailed structure of the ADown module showing its internal components.

**Figure 2 animals-15-01000-f002:**
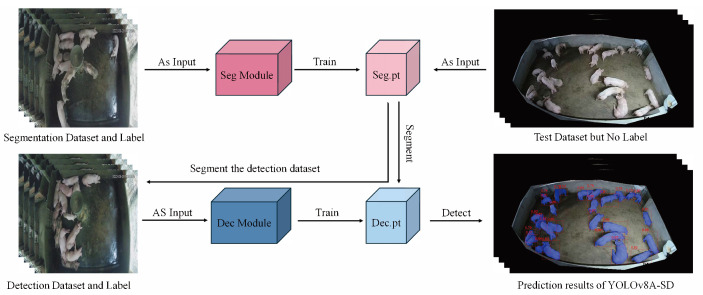
Illustrates the flowchart for the training and inference of the YOLOv8A-SD model. The Seg Module refers to the segmentation component of the model; the Dec Module denotes the detection component of the model; Seg.pt represents the model derived from segmentation training; Dec.pt is the weight file (comprising Dec-NS.pt and Dec-S.pt) generated from detection training.

**Figure 3 animals-15-01000-f003:**
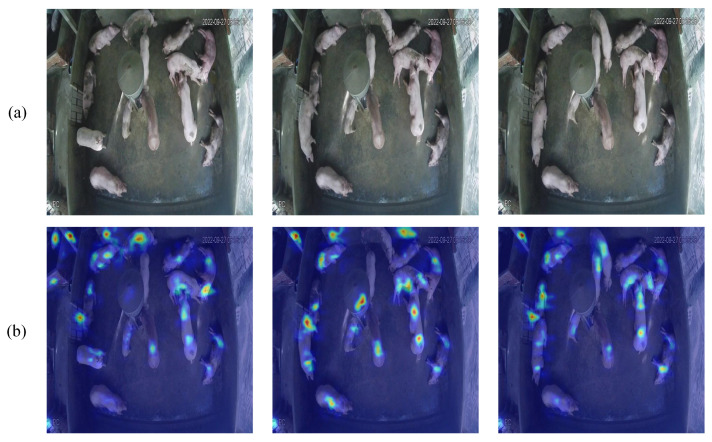
YOLOv8A-SD model Grad-CAM visualization effect: (**a**) Original image; (**b**) Grad-CAM heat map, where red and yellow regions indicate areas of high attention by the model, while blue regions represent areas of low attention.

**Figure 4 animals-15-01000-f004:**
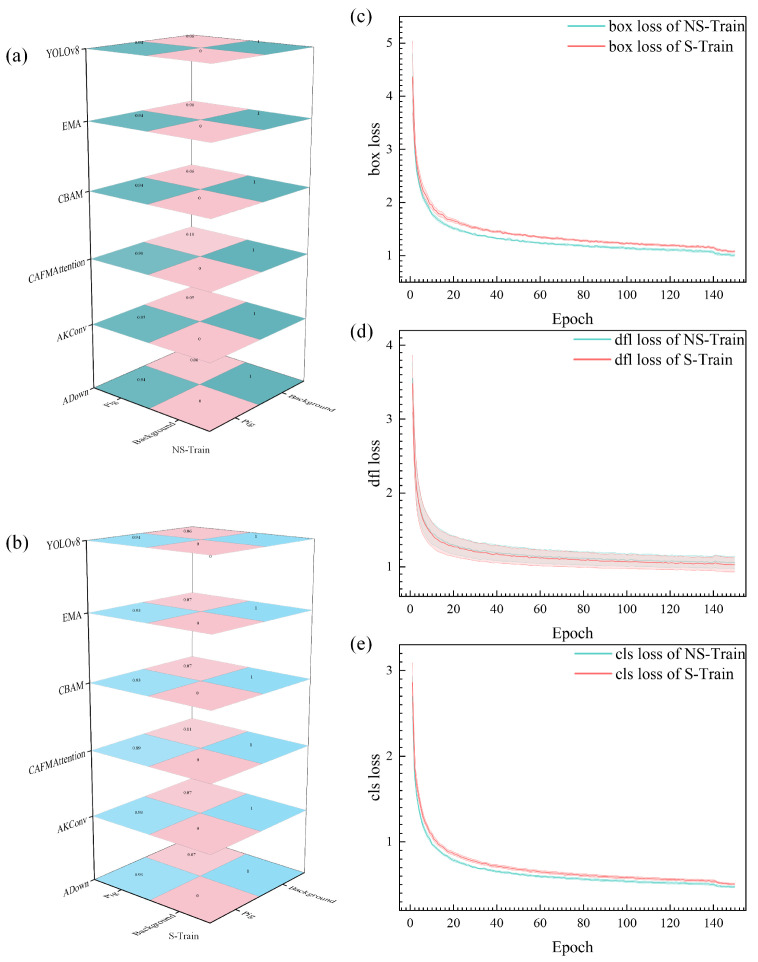
Confusion matrix and loss visualization throughout the training phase. (**a**): Confusion matrix derived from the S-Train dataset; (**b**): Confusion matrix derived from the NS-Train dataset; (**c**): Box loss comparison between S-Train (red) and NS-Train (green); (**d**): DFL loss comparison between S-Train (red) and NS-Train (green); (**e**): Classification loss comparison between S-Train (red) and NS-Train (green).

**Figure 5 animals-15-01000-f005:**
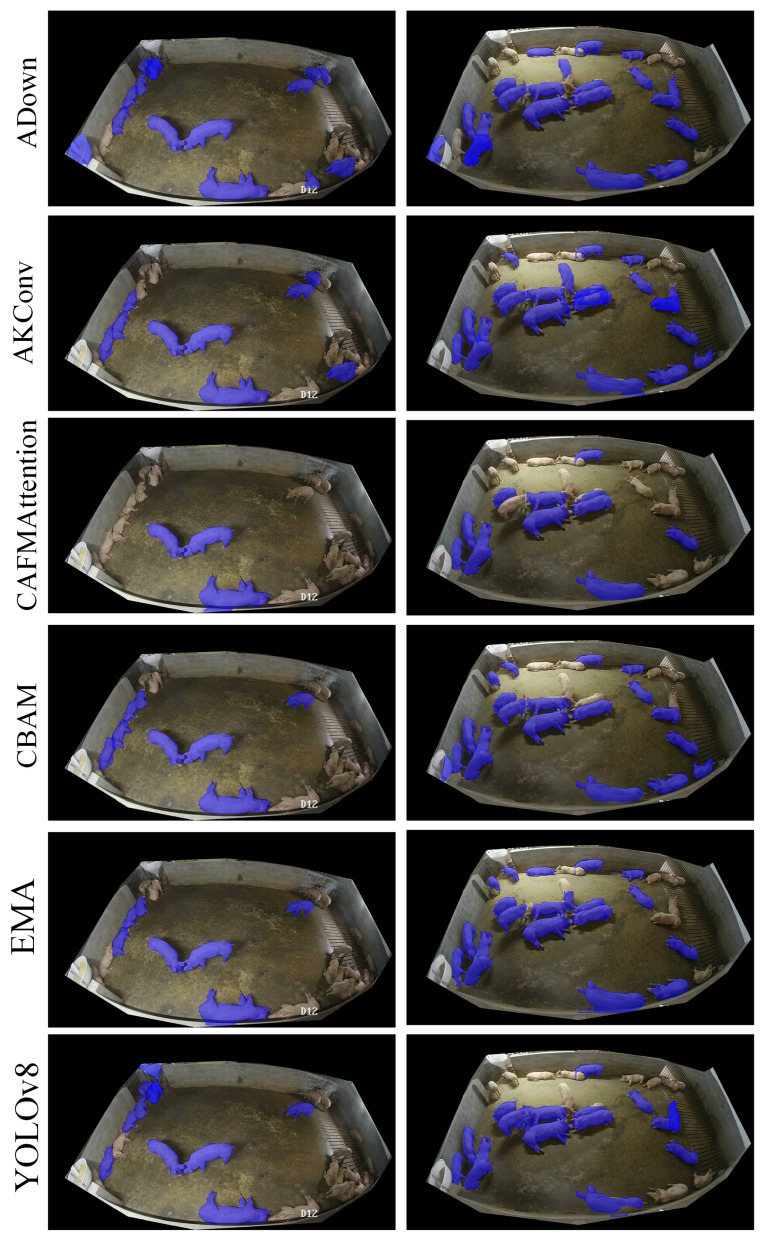
Segmentation of YOLOv8A-SD segmentation model in different scenarios. The left column shows original surveillance images of pigs in a sparse distribution, while the right column shows a clustered distribution. Each row represents a different attention mechanism’s segmentation results (from top to bottom: ADown, AKConv, CAFMAttention, CBAM, EMA, and baseline YOLOv8), with blue mask labeling visualizing the model’s instance segmentation capability.

**Figure 6 animals-15-01000-f006:**
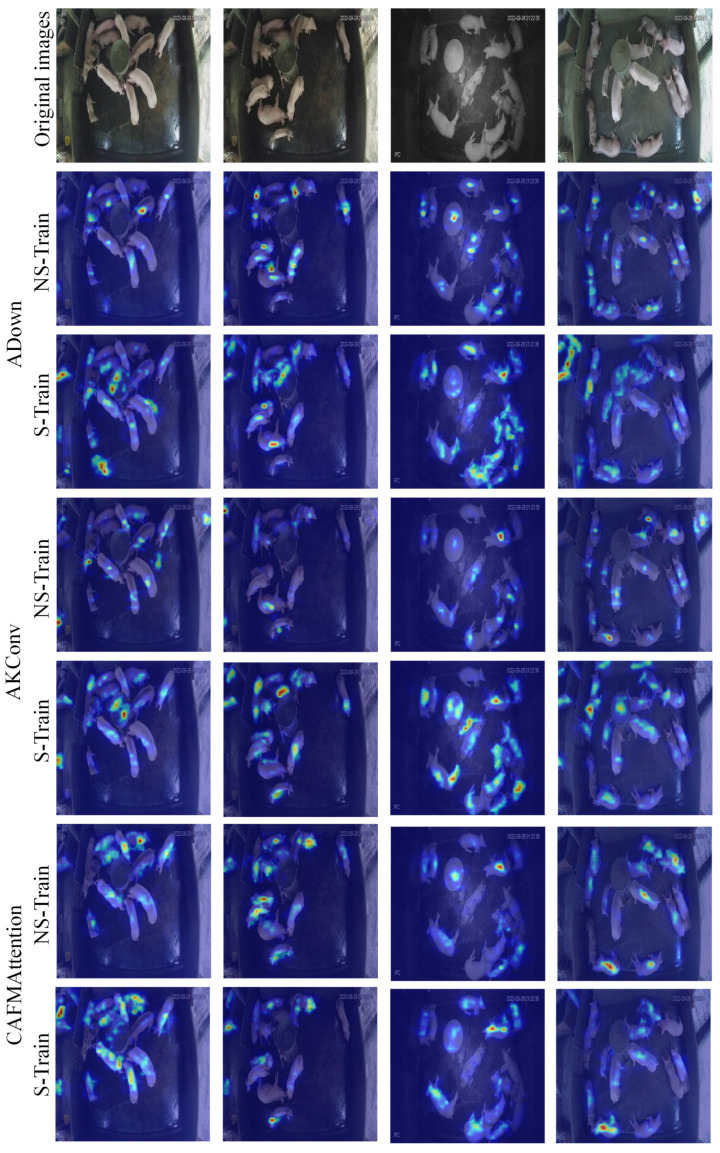
Heat map visualization of ADown, AKConv, and CAFMAttention modules. The first row shows original surveillance images, while subsequent rows display heat maps for different training modes (NS-Train and S-Train) of each module. In the heat maps, red and yellow regions indicate areas of high attention, while blue regions represent areas of low attention by the model.

**Figure 7 animals-15-01000-f007:**
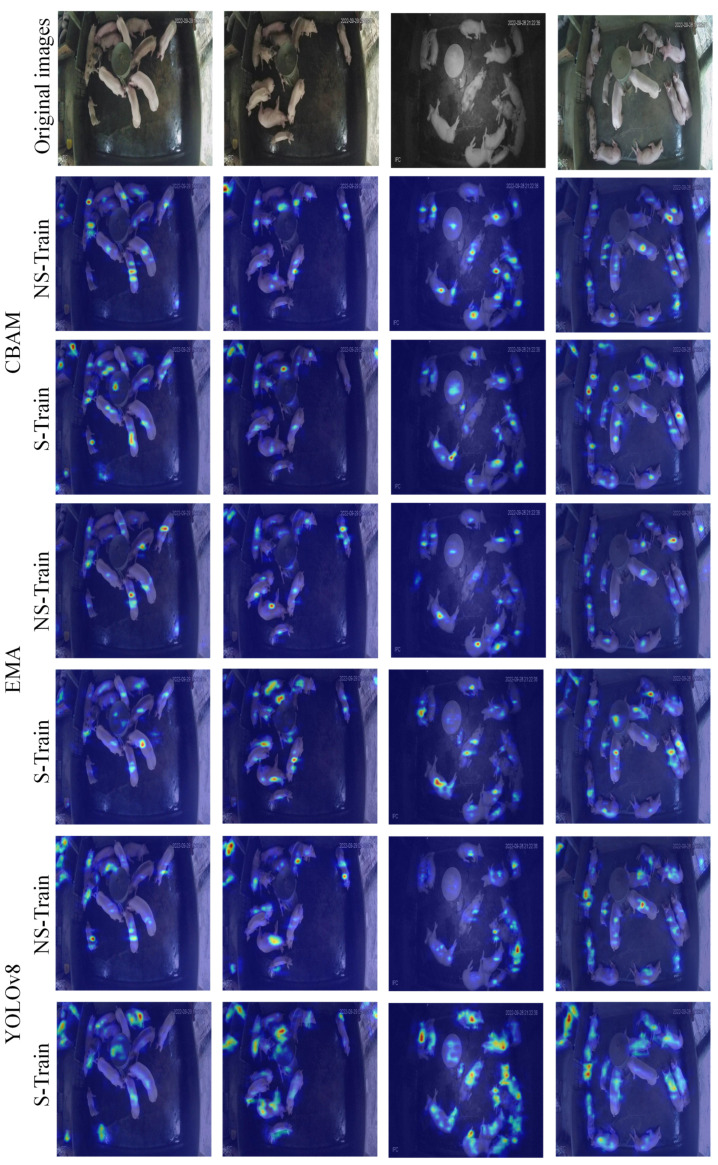
Heat map visualization of CBAM, EMA, and original YOLOv8 modules. The heat maps use the same color scheme where red and yellow indicate high attention areas and blue indicates low attention areas.

**Figure 8 animals-15-01000-f008:**
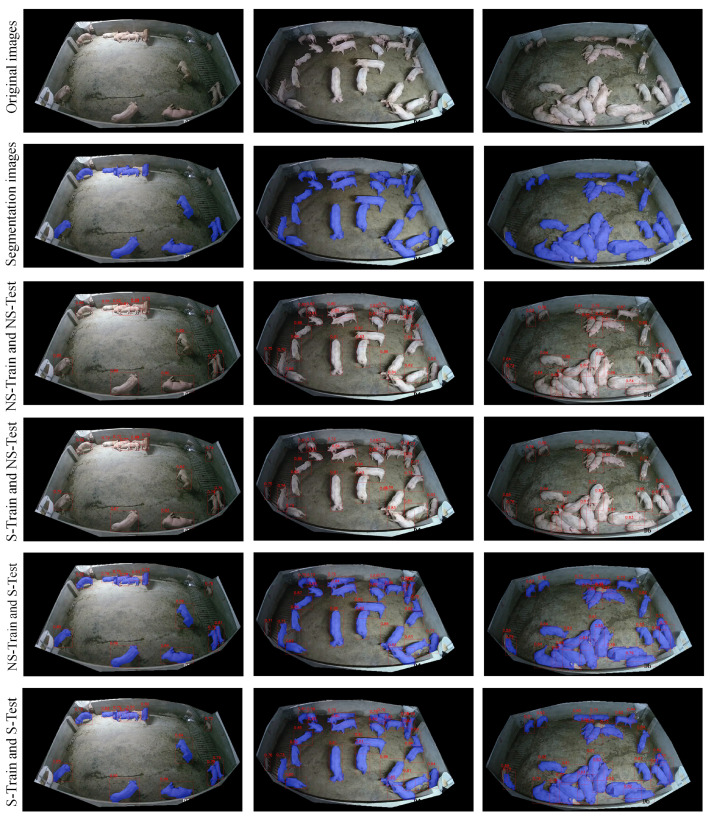
Effectiveness of different training strategies of YOLOv8a-SD in pig target detection. The first row shows original surveillance images, while subsequent rows show detection results under different training-testing combinations. Red bounding boxes indicate detection results, while blue masks represent segmentation results. NS-Test: the tested image is not segmented; S-Test: the tested image is segmented; NS-Train: the trained image is not segmented and S-Train: the trained image is segmented.

**Table 1 animals-15-01000-t001:** YOLOv8A-SD model training parameter configurations.

Experimental Parameter	Parameter Configuration
Operating System	Windows 11
GPU	NVIDIA GeForce RTX 2080Ti
CPU	Intel Core i5-12600KF
Deep Learning Framework	PyTorch 2.4.1
CUDA Version	CUDA 11.8
Python Version	Python 3.8.20
Epoch	150
Optimizer	Adam
Initial Learning rate	0.01
Learning rate Scheduler	CosineAnnealingLR

**Table 2 animals-15-01000-t002:** Evaluation of performance measures for various attention modules on the NS-Train and S-Train datasets. The initial five modules (ADown, AKConv, CAFMAttention, CBAM, EMA) constitute the attention mechanism modules, whereas None indicates the absence of supplementary modules in the original YOLOv8 model; the Precision, Recall, and mAP50 values in the table are averaged over the final ten rounds of model training.

Module	Precision (%)		Recall (%)		mAP50 (%)	
	NS-Train	S-Train	NS-Train	S-Train	NS-Train	S-Train
ADown	96.1	94.8	90.6	88.8	96.3	95.3
AKConv	95.8	95.4	90.3	88.5	96.5	95.3
CAFMAttention	95.0	94.5	87.6	87.9	93.9	94.8
CBAM	95.5	93.8	90.2	84.7	96.3	92.4
EMA	95.6	94.9	90.1	87.7	96.1	94.8
YOLOv8	95.7	95.1	90.2	89.0	96.0	95.6

**Table 3 animals-15-01000-t003:** Comparison of segmentation performance of different attention modules.

Module	Precision (%)	Recall (%)	IoU (%)	Dice (%)
	Value	Value	Value	Value
ADown	93.1	88.6	83.1	90.8
AKConv	92.2	88.9	82.7	90.5
CAFMAttention	90.8	86.4	79.5	88.6
CBAM	91.9	87.9	81.6	89.9
EMA	91.4	87.2	80.6	89.3
YOLOv8	91.7	87.4	81.0	89.5

**Table 4 animals-15-01000-t004:** Ablation study of different components in YOLOv8A-SD.

Components	Detection Metrics (%)			Segmentation Metrics (%) ^a^	
	Precision	Recall	mAP50	IoU	Dice
YOLOv8 (Baseline)	91.7	87.4	96.0	81.0	89.5
+ADown (S-Train)	95.1	89.0	95.6	83.1	90.8
+ADown (NS-Train)	96.1	90.6	96.3	-	-
Strategy					
Train mode ^b^	Test mode ^c^	Num of Pigs		Range of Conf (Avg) ^d^	
YOLOv8 (Baseline)					
NS-Train	NS-Test	25.39		0.388∼0.869 (0.741)	
NS-Train	S-Test	25.25		0.387∼0.880 (0.761)	
S-Train	NS-Test	25.81		0.371∼0.864 (0.731)	
S-Train	S-Test	26.34		0.396∼0.842 (0.714)	
YOLOv8A-SD					
NS-Train	NS-Test	25.17		0.388∼0.908 (0.767)	
NS-Train	S-Test	25.05		0.374∼0.897 (0.783)	
S-Train	NS-Test	24.35		0.414∼0.881 (0.752)	
S-Train	S-Test	25.24		0.286∼0.892 (0.740)	
Actual value		25.09		-	

^a^ Segmentation Metrics evaluate the model’s segmentation performance. The dash (-) indicates that only one segmentation experiment was conducted for the YOLOv8A-SD model. ^b^ Train mode uses a 2.8 m dataset where NS-Train means training on original images, and S-Train means training on segmentation-preprocessed images. ^c^ Test mode uses a 3.5 m dataset where NS-Test means testing on original images, and S-Test means testing on segmentation-preprocessed images. ^d^ Conf represents confidence score, and Avg in parentheses represents the average confidence score.

## Data Availability

The data presented in this study are available on request from the corresponding author.

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
