# Peer review of "YOLOv8A-SD: A Segmentation-Detection Algorithm for Overlooking Scenes in Pig Farms"

_animals, 2025, doi:10.3390/ani15071000_

Round 1
Reviewer 1 Report
Comments and Suggestions for Authors
If this is an article for the Animals journal, then a Simple Summary is also required in addition to the Abstract at the beginning. It is worth supplementing the article with this text element.
At the end of the Introduction chapter, the Authors wrote what the study presented in the article. In this case, instead of what the study presents, it would be worth writing about the aim of the presented research study. Then, in the further part of the article, it is easier to refer to the stated aim and indicate whether the Authors achieved this aim in whole or in part. Expanding on the issue of the aim of the research study, it would also be worth writing in the article what the cognitive (scientific) aim was and what the utilitarian (practical) aim of the conducted research was. After reading the entire article, I found the sentence: "The project aims to increase the design ..." at the end of the article (in the Conclusions, line 376). In my opinion, the sentence in which the aim of the project (study) was stated would be worth writing at the beginning, not at the end of the article.
It would be worth writing about the research problem that the authors undertook to solve. The research problem results from a review of the state of knowledge. The authors have reviewed the knowledge, so there is justification for formulating a research problem. In connection with the research problem, one can indicate a gap in the current state of knowledge, which the Authors want to fill with their research.
In the article, the Authors develop various versions of the YOLO model. I think it would be worth providing the full name of the YOLO acronym. Especially since the article contains many related acronyms, i.e., PDC-YOLO, PBR-YOLO, YOLO-DLHS-P, TR-YOLO, YOLOv5-KCB, YOLOv8A-SD, and other acronyms, e.g., ELAN (line 49), CA (line 59). I admit that with such many acronyms, it is difficult to figure out the differences between them and what individual features distinguish individual models, which distinguish them from other models with similar (yet different) names.
From line 68, chapter 2. Methods have been developed. I want to ask why the chapter is not titled Methods and Materials. The main idea of the research study is to develop an algorithm for handling research data, which can be treated as a methodological approach. However, obtaining this research data required the involvement of research material, i.e. animals. Therefore, in my opinion, it is justified to include not only the method but also the material in the title of the chapter.
In line 73, the Authors mentioned the camera apparatus used in the study. If research equipment is used (in the study), it would be good to provide more details about this equipment (e.g., manufacturer and technical details). The same line also states, “… camera apparatus positioned at a fixed height …”. If necessary, it would be good to provide the height to repeat the experiment based on the details. I realize that the primary goal of the research study was to improve the method/algorithm for handling data. Still, it seems worth devoting more space and attention to the animals and their housing conditions. Without the participation of animals and considering their behavior, it would be difficult to conduct the planned research study. Therefore, I suggest providing more details about the pigs in the study. What breed were the pigs? What was the average weight of the animals in the herd (pen) during the study period? What were the conditions of keeping the pigs, i.e., what was the herd density (how many animals per square meter, or rather how many square meters per animal), and on what type of surface (floor) were the pigs kept? Did the pigs have constant access to feed and water? These final issues are related to animal welfare, which should be maintained and mentioned when presenting the conditions for conducting observations with animals. The article's introduction used the word " welfare " only once. I think it would be worth expanding a bit on the issue of pig welfare, which may translate into their behavior, especially since images that account for the details of pig behavior are the subject of improving the algorithm for handling the recorded material.
If three equations (1), (2), and (3) were given, in what units are the calculation results determined? You can guess this, but it is also worth writing it in the text describing the patterns on page 5.
In the caption to Figure 5, it would be worth writing what details were presented in the figures marked (a) and what in the figures (b).
The article includes 31 citations. The last citation is on page 2, in the Introduction chapter. Chapter 3 is titled Results and Discussion. Why does the discussion in this chapter only consider the results of their research? If, in the Introduction, the Authors included many YOLO methods with modifications of PDC, PBR, etc., then it might be worth comparing the results of the presented research study with the results of research on YOLO with modifications.
I want to ask in what practical way the owner of the animal herd can use the research study results and what benefits can be achieved on the farm. What skills must a farmer have to use the proposed procedure algorithm to achieve higher efficiency of work tasks in pig production?
Reviewer 2 Report
Comments and Suggestions for Authors
The research is focused on the detection of pigs through computer vision, but the authors do not refer to any detailed application context.
The definition of the study objectives should start from the analysis of specific PLF challege(s). Then, computer vision can be considered to that end and the study can be motivated.
Moreover, it is necessary to specify if the method is capable of identifying individual animals and give reason of the answer.
Comments on the Quality of English LanguageThe use of the term "carcass" referred to the body of a living pig looks inappropiate.
Reviewer 3 Report
Comments and Suggestions for Authors
This manuscript proposes an improved object detection and segmentation algorithm by integrating the ADown attention mechanism and a segmentation-detection dual-task strategy. However, the following issues need to be addressed:
- Some references in the introduction are outdated. It is recommended to include updated literature on the YOLO series and Transformer-based object detection methods. The following reference may be useful: 10.3390/ani14213059.
- The object detection model appears to lack an independent test set.
- The manuscript lacks essential details about the training environment, such as the operating system, deep learning framework version, and CUDA version.
- The evaluation metrics only cover object detection, but there are no metrics for semantic segmentation, such as IoU and Dice coefficient.
- Figure 4 has readability issues, such as poor color contrast, unclear labels, and missing numerical annotations.
- The edges of images in Figures 5 and 6 are blurry.
- The manuscript lacks a comparative analysis between the proposed method and the original YOLOv8 in the semantic segmentation task.
- The manuscript introduces the ADown attention mechanism and segmentation-detection dual-task strategy but lacks ablation experiments for key modules.
- The comparison between NS-Train (original image training) and S-Train (segmentation-preprocessed training) is unclear, and the reason for the performance drop in S-Train has not been thoroughly analyzed.
- The manuscript does not provide a detailed analysis of error cases (e.g., false positives and false negatives), and it lacks an error analysis and failure case discussion.
Round 2
Reviewer 1 Report
Comments and Suggestions for Authors
Thank you for the answers given in the issues taken in the article review. Thank you for taking my comments and introducing changes and supplementations in the article.
Reviewer 2 Report
Comments and Suggestions for Authors
The manuscript has been properly revised.
Reviewer 3 Report
Comments and Suggestions for Authors
The author has solved the previous problem well, and this article is ready for publication.